# Associations between WASH-related violence and depressive symptoms in adolescent girls and young women in South Africa (HPTN 068): a cross-sectional analysis

Ruvani T Jayaweera [1,2] Dana Goin,[3] Rhian Twine,[4] Torsten B Neilands,[5] Ryan G Wagner,[4] Sheri A Lippman,[4,5] Kathleen Kahn,[4,6] Audrey Pettifor,[4,7] Jennifer Ahern[2]

**Correspondence to**
Dr Ruvani T Jayaweera;
rjayaweera@ibisreproductive
health.org

## ABSTRACT

**Objective** There is a lack of research on experiences of WASH-related violence. This study aims to quantify the association between experience or worry of violence when using the toilet or collecting water and depressive symptoms among a cohort of young women in South Africa.

**Methods** Data are from visit 3 of the HPTN 068 cohort of adolescent girls in rural Mpumalanga Province, South Africa. Participants (n=1798) included in this analysis were aged 13–21 at baseline. Lifetime experience of violence or fear of violence when using the toilet and collecting water was collected by self-report; depressive symptoms in the past week were measured using the Center for Epidemiological Studies Depression Scale (CES-D). We used G-computation to calculate the prevalence difference (PD) and prevalence ratio of depression (CES-D score >15) associated with each domain of violence, controlling for baseline covariates.

**Findings** A total of 15.1% of respondents reported experiencing violence when using the toilet; 17.1% reported experiencing violence when collecting water and 26.7% reported depression. In adjusted models, those who reported experiencing violence when using the toilet had an 18.1% higher prevalence of depression (95% CI: 11.6% to 24.4%) than those who did not experience violence when using the toilet. Adjusted prevalence of depression was also higher among those who reported violence when collecting water (PD 11.9%, 95% CI: 6.7% to 17.2%), and who worried about violence when using the toilet (PD 12.8%, 95% CI: 7.9% to 19.8%), as compared with those who did not report these experiences. Worrying about violence when collecting water was not associated with depression after adjusting for covariates.

**Conclusion** Experience of WASH-related violence is common among young women in rural South Africa, and experience or worry of experiencing violence is associated with higher prevalence of depressive symptoms.

**Trial registration number** NCT01233531; Post-results.

## STRENGTHS AND LIMITATIONS OF THIS STUDY

⇒ This is the first study to quantitatively assess the effect of experiencing violence or fear of experiencing violence on mental health.
⇒ Strengths include a large study design, and detailed questions on direct experiences of WASH-related violence and the type of WASH-related activity where the violence occurred.
⇒ Additional strengths include a rigorous analytic approach to quantify the absolute marginal difference in prevalence of depression, adjusting for covariates.
⇒ Limitations are the cross-sectional nature of the study, and possible under-reporting of violence as linked to sanitation experiences given the sensitivities around experience of violence, particularly sexual violence.

## INTRODUCTION

Despite long-standing global efforts, approximately two billion people around the world lack access to safe water and 3.6 billion lack access to safely managed sanitation.[1] Much of the existing public health research on water, sanitation, and hygiene (WASH) has focused on pathogen-related risks to health or long-term health impacts of infectious diseases.[2] However, this does not capture the breadth of health consequences that arise due to challenges accessing and meeting daily WASH needs. While recent studies have noted the increased vulnerability to violence that women and girls face when meeting their daily WASH needs,[3 4] the experience of physical or sexual violence when travelling to or using water and sanitation facilities (WASH-related violence) remains an under-examined issue. This paper seeks to quantify the burden of exposure to WASH-related violence and explore its relationship to mental health in a cohort of young women and adolescent girls in South Africa.

Around the world, women and girls hold primary responsibility for meeting their household's daily water needs.[3] Coupled with the additional challenge of managing menstruation, and additional stigma faced by women for open defecation, women and girls are most impacted by lack of access to WASH services.[4] A literature review highlighted many examples from humanitarian case studies and practitioner reports that document the heightened risk of sexual violence faced when using communal latrines or practicing open defecation, particularly at night, as well as the heightened risk of sexual violence when travelling to collect water.[4] In addition to the risk of sexual violence, experience of physical violence (eg, fights over resources when queuing for water, intimate partner violence over inability to meet the household's daily water needs) was also identified as a significant public health issue.[4]

A growing body of research has further elaborated the ways in which shared water and sanitation facilities are a potential site of violence against women.[5] Qualitative research from Uganda, Kenya, and India have found that fear of sexual or physical assault and lack of privacy are driving factors in women's sanitation practices and an important source of anxiety and stress.[3 6–8] Quantitative evidence on the linkages between WASH and experiences of violence from India,[9 10] South Africa,[11] and Kenya[12] have found that women who use open defecation or shared toilet facilities have higher odds of reporting non-partner sexual violence when compared with women with a household toilet. An analysis of Demographic and Health Survey data from 25 African countries found that unimproved water and sanitation facilities were associated with higher odds of experiencing intimate partner violence.[13] An ecological study found a positive association between the number of sanitation facilities and the rate of reported sexual assaults in a township in South Africa.[14] However, none of these studies assessed whether women experienced violence while they were accessing or using WASH facilities.

While violence is a critical public health and human rights issue on its own, there are also consequent effects of violence on other aspects of health and well-being. While the immediate health impacts of poor access to water and sanitation facilities, such as exposure to water-borne pathogens, vaginal and urinary tract infections, malnutrition, dehydration, and hunger[6] are well-studied, research linking WASH-related violence to additional downstream health impacts is limited. However, new measures of water[15] and sanitation[16] insecurity include concerns about violence as a domain in these measures, and there is a growing body of research linking water and sanitation insecurity with poor mental health outcomes.[17–19] However, to our knowledge, no studies have quantitatively explored the relationship between direct experiences of violence experienced while accessing or using WASH facilities on mental health.

Given the lack of evidence on direct experiences with violence when accessing water or sanitation facilities and the potential for downstream health impacts, this paper seeks to quantify the association between experience or fear of violence when using the toilet or collecting water with depressive symptoms among a cohort of adolescent girls and young women in South Africa.

## METHODS

### Study design

Data for this study are from the HIV Prevention Trials Network (HPTN) 068 cohort of adolescent girls and young women in rural Mpumalanga Province, South Africa, a longitudinal cohort established in 2012 to estimate the effect of cash transfers, conditional on staying in school, on HIV incidence. Participants were eligible if they were between the ages of 13 and 20 years, enrolled in grades 8–11 at a participating public school, unmarried, not pregnant at the time of enrollment, able to read, had parents or guardians able to open a bank account, and resided in the Medical Research Council/Wits University Agincourt Health and Socio-Demographic Surveillance System (AHDSS) study site. The AHDSS study site is in a rural area of Mpumalanga Province, South Africa that is characterised by high HIV prevalence, high poverty, and migration for work.[20] Most households lack access to piped water in their dwellings, and sanitation is rudimentary.[20]

All households with eligible adolescent girls and young women in the study area were recruited. A total of 2533 participants enrolled and were followed annually for up to 4 years. At each study visit, participants completed interviewer-administered surveys on a wide variety of domains that included economic activities, health behaviours, health knowledge, and attitudes towards social norms. Sensitive items, such as sexual behaviour and mental health, were completed by the participants themselves via Audio Computer Assisted Self Interview (ACASI), where participants listen to questions and response categories through headphones and select their responses. Prior research has found higher reporting of sensitive issues via ACASI as compared with interviewer-administered surveys.[21] Participants' heads of households completed surveys about household composition and wealth at each visit. Full details on the study recruitment and procedures, including a full description of the sample[22] and primary trial outcomes,[23] have been published elsewhere.

### Measures

The primary outcome of interest for this study is depression, assessed using the Center for Epidemiological Studies Depression Scale (CES-D).[24] The CES-D is a 20-item measure that assesses symptoms of depression over the past week, with frequency of experiencing each symptom as rarely/none of the time, some of/a little of the time, occasional or a moderate amount of time, or all of the time. Scores can range from 0 to 60; in keeping with the literature, we used a cut-off of 16 or greater as an indicator of depression.[25]

Our exposures of interest are WASH-related violence, assessed across four domains: experience of violence when collecting water, experience of violence when using the toilet, fear of violence when collecting water and fear of violence when using the toilet. Direct experience of violence was assessed by the following question: 'How often have you experienced violence when collecting water?' and fear of experiencing violence was assessed by the following question: 'Do you ever feel concerned or worried about experiencing violence when using the toilet?' Participants were categorised as being exposed to experience of violence or fear of violence if they responded 'Just a few times', 'Regularly/about once a week' or 'Every day', as opposed to 'Never'. Though both direct experiences with violence and fear of experiencing violence were assessed at the same time point, based on the wording of the question, we assume that direct experience of violence precedes fear of experiencing violence.

We used a directed-acyclic graph to identify a minimally sufficient set of literature-based confounders available in our study. We identified the following sociodemographic covariates of interest: age at time of survey,[11 12 26] maternal and paternal education,[26] orphan status,[26] household food insecurity in the past 30 days,[11 18 27] decile of household capita consumption[12 26] and any negative events experienced by the household in the past 12 months[18 27] (assessed via the household survey). Negative events reported in the household survey included experiences such as death or serious illness of a household member, loss of livestock or crop failure, job loss or loss of government grants or loss or destruction of property. We also controlled for toilet type[9 11] and household water source.[18]

## Analysis

Data for this analysis are drawn from visit 3. We limited our analysis to this time point as only 35.5% of the enrolled sample participated in visit 4 given planned study exit due to graduation from high school. While this data is cross-sectional, experiences or fear of violence was assessed as a lifetime measure, and we assume that those experiences precede the depression measure, which evaluates depression symptoms experienced in the past week. We also assume that experience of violence precedes fear of violence.

Records with missing data on parental education were treated as a separate category in analysis; missing data on household food insecurity (n=7), decile of total household per capita expenditure (n=8) and orphan status (n=17) were directly extracted from prior household survey visits.

We used G-computation to calculate the predicted marginal prevalence difference of depression and the predicted marginal prevalence ratio of depression associated with each individual domain of WASH-related violence. All models adjusted for age at time of visit, maternal education, paternal education, orphan status, household food insecurity, decile of household capita

consumption, negative household experiences, and trial arm, and accounted for clustering by village by using the nonparametric cluster bootstrap to calculate 95% Wald-type percentile-based CIs from 500 resamples. Models assessing fear of violence when collecting water and fear of violence when using the toilet additionally adjusted for prior experience of violence when collecting water and when using the toilet, respectively. Models assessing experience or fear of violence when collecting water additionally adjusted for household water source; models assessing experience or fear of violence when using the toilet additionally adjusted for household toilet type. All analyses were performed using Stata V.15.[28]

## Patient and public involvement

All studies, including the HPTN 068 trial, conducted in the AHDSS study site receive permission to undertake research activities from a forum comprised of community and village leaders. Findings from the main trial were communicated to the community via community meetings and factsheets. Additional details on community involvement are available via the study site's Public Engagement Office.[29]

## RESULTS

A total of 1870 participants completed a survey at visit 3. Observations were excluded from analysis if they did not complete the CES-D items or the WASH-related violence questions (n=61, 3.3%). Participants were additionally excluded as they did not have any data on orphan status or household data on toilet type and water source (n=11, 0.6%) at any time point, yielding a final analytic sample of 1798 observations.

Table 1 displays the sociodemographic characteristics of participants at visit 3. Participants ranged in age from 14 to 22 years; most (79.1%) were between the ages of 16 and 19 at visit 3. One-third (33.6%) of participants had at least one deceased parent, and 7.9% lived in households that reported food insecurity in the past 30 days. The overall prevalence of depression in the sample at visit 3 was 26.7%. A total of 19.4% of participants reported ever being worried about experiencing violence when using the toilet; 16.5% reported being worried about experiencing violence when collecting water. Directly experiencing violence when using the toilet was reported by 15.1% of participants; experiencing violence when collecting water was reported by 17.1%. A combined total of 26.2% reported directly experiencing violence when using the toilet or collecting water, and 29.3% reported ever being worried about experiencing violence when using the toilet or collecting water. Age, water source and toilet type were all associated with depressive symptoms in Pearson $\chi^2$ tests of independence adjusted for clustering by village; age and water source were also associated with any experience of violence with WASH-related violence, and age was associated with fear of WASH-related violence.

Table 1 Demographic characteristics and prevalence of depression, experience with WASH-related violence and fear of WASH-related violence (n=1798)

| Covariate | n | % | Depressive symptoms* (%) | Direct experience of WASH violence* (%) | Fear of WASH violence* (%) |
|---|---|---|---|---|---|
| Age at visit† | | | | | |
| 14 | 2 | 0.1 | 0 | 50 | 50 |
| 15 | 259 | 14.4 | 17.4 | 21.2 | 19.7 |
| 16 | 449 | 25.0 | 24.7 | 26.1 | 27.2 |
| 17 | 459 | 25.5 | 25.9 | 24.4 | 29.2 |
| 18 | 341 | 19.0 | 32.6 | 27.3 | 33.7 |
| 19 | 173 | 9.6 | 28.3 | 27.7 | 34.7 |
| 20 | 70 | 3.9 | 37.1 | 37.1 | 37.1 |
| 21 | 30 | 1.7 | 36.7 | 33.3 | 30.0 |
| 22 | 15 | 0.8 | 53.3 | 60.0 | 53.3 |
| Orphan status | | | | | |
| One parent deceased | 493 | 27.4 | 27.8 | 24.5 | 26.2 |
| Two parents deceased | 112 | 6.2 | 26.8 | 25.0 | 27.7 |
| Maternal education | | | | | |
| No school | 288 | 16.0 | 27.8 | 25.3 | 28.8 |
| Some primary | 320 | 17.8 | 29.7 | 28.7 | 31.3 |
| Completed primary | 79 | 4.4 | 25.3 | 31.6 | 31.6 |
| Some high school | 514 | 28.6 | 26.5 | 26.1 | 30.7 |
| Completed high school | 386 | 21.5 | 23.8 | 25.4 | 27.7 |
| University or vocational | 57 | 3.2 | 15.8 | 17.5 | 22.8 |
| Unknown/missing | 154 | 8.6 | 31.2 | 25.3 | 26.0 |
| Paternal education | | | | | |
| No school | 295 | 16.4 | 28.4 | 26.8 | 31.2 |
| Some primary | 246 | 13.7 | 28.9 | 30.5 | 36.2 |
| Completed primary | 80 | 4.4 | 26.3 | 20.0 | 23.8 |
| Some high school | 339 | 18.9 | 22.4 | 26.5 | 28.6 |
| Completed high school | 425 | 23.6 | 26.6 | 25.4 | 28.2 |
| University or vocational | 71 | 3.9 | 21.1 | 21.1 | 21.1 |
| Unknown/missing | 342 | 19.0 | 29.2 | 25.7 | 27.5 |
| Household food insecurity | | | | | |
| No | 1656 | 92.1 | 26.3 | 26.1 | 29.2 |
| Yes | 142 | 7.9 | 31.0 | 26.8 | 29.3 |
| Decile of total household per capita expenditures | | | | | |
| 1 | 181 | 10.1 | 27.1 | 24.9 | 30.4 |
| 2 | 178 | 9.9 | 29.8 | 28.7 | 36.0 |
| 3 | 183 | 10.2 | 27.9 | 29.0 | 29.5 |
| 4 | 176 | 9.8 | 31.3 | 25.6 | 30.1 |
| 5 | 176 | 9.8 | 27.3 | 24.4 | 31.3 |
| 6 | 182 | 10.1 | 23.6 | 22.0 | 26.9 |
| 7 | 177 | 9.8 | 24.3 | 23.2 | 23.2 |
| 8 | 181 | 10.1 | 27.6 | 25.4 | 23.8 |
| 9 | 183 | 10.2 | 25.7 | 30.1 | 31.1 |
| 10 | 181 | 10.1 | 22.7 | 28.7 | 30.4 |

Continued

**Table 1** Continued

| Covariate | n | % | Depressive symptoms* (%) | Direct experience of WASH violence* (%) | Fear of WASH violence* (%) |
|---|---|---|---|---|---|
| Household experienced any recent negative event | | | | | |
| No | 1463 | 81.4 | 26.0 | 26.2 | 29.2 |
| Yes | 335 | 18.6 | 29.6 | 26.3 | 29.6 |
| Household water source‡ | | | | | |
| Piped water | 1188 | 66.1 | 24.7 | 23.6 | 27.4 |
| Public tap/standpipe | 460 | 25.6 | 29.8 | 32.8 | 34.8 |
| Rain/surface water | 15 | 0.8 | 20.0 | 33.3 | 26.7 |
| Tanker truck | 88 | 4.9 | 33.0 | 25 | 25.0 |
| Well (tube, dug, borehole) | 47 | 2.6 | 38.3 | 27.7 | 29.8 |
| Household toilet type§ | | | | | |
| Flush or pour toilet | 106 | 5.9 | 14.2 | 24.5 | 33.0 |
| No facility/open defecation | 88 | 4.9 | 25.0 | 27.3 | 29.5 |
| Pit latrine | 1604 | 89.2 | 27.6 | 26.2 | 29.0 |
| Trial arm¶ | | | | | |
| 1 | 868 | 48.3 | 27.3 | 28.5 | 31.8 |
| 2 | 930 | 51.7 | 25.9 | 24.1 | 26.9 |
| **Total** | **1798** | **100** | **26.7** | **26.2** | **29.3** |

*Proportion in each sociodemographic subcategory reporting depressive symptoms, any direct experience of WASH-related violence (while collecting water or using the toilet), or any worry of WASH-related violence (while collecting water or using the toilet).
† p=0.001 (age and depressive symptoms), p=0.017 (age and any experience of violence), p=0.017 (age and any fear of violence).
‡ p=0.027 (water source and depressive symptoms), p=0.039 (water source and any experience of violence).
§ p=0.010 (toilet type and depressive symptoms).
¶ p=0.014 (trial arm and any experience of violence), p=0.036 (trial arm and any fear of violence).
WASH, water, sanitation and hygiene.

G-computation results from estimating the absolute and relative association of each exposure on the prevalence of depression (CES-D ≥16) can be found in table 2. After adjusting for covariates, experiencing violence when using the toilet was associated with a 1.76 times higher prevalence of depression (95% CI: 1.47 to 2.05) and experiencing violence when collecting water was associated with a 1.48 times higher prevalence of depression (95% CI: 1.26 to 1.77). While fear of experiencing violence when collecting water was not associated with a

**Table 2** Adjusted prevalence ratios and prevalence differences for depression across four WASH-related violence exposures (n=1798)

| Exposure* | Prevalence ratio (95% CI)† | Prevalence difference† |
|---|---|---|
| Experienced violence when using the toilet‡ | 1.76 (1.47 to 2.05) | 18.1% (11.6% to 24.4%) |
| Experienced violence when collecting water§ | 1.48 (1.26 to 1.77) | 11.9% (6.7% to 17.2%) |
| Worried of experiencing violence when using the toilet¶ | 1.53 (1.31 to 1.92) | 12.8% (7.9% to 19.8%) |
| Worried of experiencing violence when collecting water** | 1.05 (0.81 to 1.44) | 1.4% (−5.3% to 10.9%) |

*Reference group for each exposure is NO experience with violence (or NO worry of experiencing violence).
†G-computation estimates are from four separate logistic regression models adjusting for covariates, with 95% CI's calculated from the non-parametric cluster bootstrap.
‡Model adjusts for age at time of visit, maternal education, paternal education, orphan status, household food insecurity, decile of household capita consumption, household experience of any negative event, trial arm and toilet type.
§Model adjusts for age at time of visit, maternal education, paternal education, orphan status, household food insecurity, decile of household capita consumption, household experience of any negative event, trial arm and water source.
¶Model adjusts for age at time of visit, maternal education, paternal education, orphan status, household food insecurity, decile of household capita consumption, household experience of any negative event, trial arm, toilet type and experience of violence when using the toilet.
**Model adjusts for age at time of visit, maternal education, paternal education, orphan status, household food insecurity, decile of household capita consumption, household experience of any negative event, trial arm, water source and experience of violence when collecting water.
WASH, water, sanitation and hygiene.

higher prevalence of depression (prevalence ratio: 1.05, 95% CI: 0.81 to 1.44), fear of experiencing violence when using the toilet was associated with a 1.53 (95% CI: 1.31 to 1.92) times higher prevalence of depression, after controlling for covariates (including prior experience of violence when using the toilet).

WASH-related violence was also associated with higher prevalence of depression on the absolute scale for experience of violence when using the toilet and collecting water, and fear of experiencing violence when using the toilet. After adjusting for covariates, experiencing violence when using the toilet (prevalence difference (PD): 18.1%; 95% CI: 11.6% to 24.4%), experiencing violence when collecting water (PD: 11.9%; 95% CI: 6.7% to 17.2%), and fear of experiencing violence when using the toilet (PD: 12.8%; 95% CI: 7.9% to 19.8%) were all associated with higher prevalence of depression. In other words, the prevalence of depression among those who reported experience with violence when using the toilet was 18.1% points higher on the absolute scale than the prevalence of depression among those who did not report any experience with violence when using the toilet, controlling for covariates.

## DISCUSSION

This study is among the first to consider how direct experience of WASH-related violence may affect mental health. Directly experiencing violence when using the toilet or collecting water was associated with substantially higher prevalence of depression; additionally, the fear of experiencing violence when using the toilet, even when controlling for prior experience of violence when using the toilet, was also associated with higher prevalence of depression. This analysis builds on previous qualitative work and practitioner reports, and identifies water and sanitation facilities as a structural and built environment determinant of violence against women and poor mental health outcomes.

While no studies have quantitatively assessed the burden of WASH-related violence, a recent study using Demographic and Health Survey data from 20 countries in Africa found that 6.2% of women reported non-partner sexual or physical violence in the previous 12 months[30]; a systematic review from 2014 estimated that the lifetime prevalence of experiencing sexual or physical violence from a non-partner is as high as 20% for women living in central Africa.[31] Given the relatively high prevalence of experiencing violence when using the toilet and collecting water in this sample (15.1% and 17.1%, respectively), our analysis suggests that water and sanitation facilities may play a substantial role in the experience of non-partner violence and may be an important site of prevention. Indeed, much of the focus on prevention of violence against women is centred on individual, household, and community-level risk factors; our findings further support calls made by others[30] to consider the role of the built environment, and specifically access to

water and sanitation facilities, as an important factor that may influence non-partner violence against women.

Additionally, our finding that direct experiences of violence when collecting water or using the toilet is associated with depression are aligned with the broader literature on the links between gender-based violence (perpetrated by intimate partners or non-partners) and depression. While the majority of the literature on mental health and gender-based violence focuses on intimate-partner violence,[32] what limited evidence does exist on non-partner violence has found that non-partner sexual violence is linked to two-to-three fold increases in odds of depressive symptoms.[33–35] Violence against women is an important human rights issue and has been well documented as having substantial mental health impacts.[36] Depressive symptoms have been linked to worse health and socioeconomic outcomes, including HIV incidence, in this cohort.[26] Thus, addressing determinants of depression related to the built environment, such as access to WASH facilities and consequent exposure to WASH-related violence, may help improve mental health outcomes as well as other forms of well-being among adolescent girls and young women. Our findings build on this literature and specifically identify the experience of physical and sexual violence in the context of meeting WASH needs as having the potential to impact mental health.

Beyond direct experiences of violence, findings in the WASH literature, detailed below, have highlighted the role that fear or worry about experiencing violence—which itself is a form of violence—may play in poor mental health outcomes. Constant stress and worry around the essential and daily activities of collecting water and using the toilet, as well as mitigating actions to minimise perceived exposure to violence, may be relevant contributors to psychosocial stress[17–19 37] which is associated with many negative health outcomes, including depression.[38] Studies globally have highlighted how navigating access to sanitation facilities contributes to psychosocial stress through fears of experiencing physical or sexual violence, particularly when needing to use shared latrines or open defecation at night.[6 7 37 39 40] Findings from this study corroborate what has been found in these studies; and quantify the substantial extent of concerns and worry about experiencing violence in the context of WASH in women's lives, even after controlling for prior experience of violence, and its consequent impact on mental health.

This study is not without limitations. Given the sensitivities around experience of violence, particularly sexual violence, there is likely under-reporting of violence as linked to sanitation experiences.[41] However, the ACASI format of the survey was used to minimise underreporting.[21] An additional limitation is the cross-sectional nature of the study: while we assume a temporal relationship between exposure to WASH-related violence and reported symptoms of depression, it is possible that some individuals had a recent experience with WASH-related violence that overlapped with the time period in which

depressive symptoms were assessed. The 20-item CESD-20 has been validated among students in South Africa,[42] though it has not been validated in this specific population. Furthermore, it is possible that the selected sociodemographic covariates do not fully control for confounding between our exposures and outcomes of interest; for example, poor parental physical and mental health, which were not recorded in this study, have been shown to be associated with higher depression among adolescents, and may also play a role in adolescent exposure to WASH-related violence via impacting the role of the adolescent in water collection for the household. While we were unable to control for distance to toilet or water source, we hypothesise that the potential confounding effects of these variables are at least partially accounted for by toilet type and water source. Additionally, school enrollment has been found to be protective against depression among youth in Africa[43] and those who face greater challenges in meeting their WASH-related needs may also be less likely to be enrolled in school; given this study is drawn from a cohort of young women enrolled in school, those who are depressed and experience WASH-violence may be underrepresented in this study, and the potential relationship between WASH-related violence and depression may be much higher. Finally, we lacked data on the location of where direct experiences of WASH-related violence or worries about WASH-related violence took place. Future studies should explore in greater detail whether experiences with WASH-related violence take place near their homes, while at school, or in other places in order to identify key sites for intervention.

The research to date demonstrates the importance of centering the needs and concerns of women when designing and implementing WASH interventions. Though outside the scope of this paper, the needs of additional groups that may face further marginalisation or risk of violence should also be considered, such as elderly people, as well as transgender and gender expansive people who likely face greater threat of violence. Additional research is warranted to better understand the scope of these individuals' experiences in order to appropriately direct resources for prevention.

This study contributes to our understanding of the scope of WASH-related violence experienced by adolescent girls and young women in Mpumalanga, South Africa, and is the first quantitative study to our knowledge that documents the relationship between direct experiences of WASH-related violence and mental health. Experience of WASH-related violence is common among young women in rural South Africa, and is associated with substantially higher prevalence of depressive symptoms. Additional research to explore the experience of violence faced by women and girls when collecting water or using the toilet is needed.

**Author affiliations**
[1]Ibis Reproductive Health, Oakland, California, USA
[2]Division of Epidemiology, School of Public Health, University of California Berkeley, Berkeley, California, USA
[3]Program on Reproductive Health and the Environment, Department of Obstetrics, Gynecology, and Reproductive Sciences, University of California San Francisco, San Francisco, California, USA
[4]MRC/Wits Rural Public Health and Health Transitions Research Unit (Agincourt), Faculty Health Sciences, School of Public Health, University of the Witwatersrand, Johannesburg, South Africa
[5]Department of Medicine, University of California San Francisco, San Francisco, California, USA
[6]Department of Public Health and Clinical Medicine, Umeå Centre for Global Health Research, Umeå University, Umeå, Sweden
[7]Department of Epidemiology, Gillings School of Public Health, University of North Carolina at Chapel Hill, Chapel Hill, North Carolina, USA

**Acknowledgements** We thank the HIV Prevention Trials Network 068 study team and all trial participants.

**Contributors** AP, RT, RGW, KK, JA and SAL contributed to the overall study conceptualisation, design, and implementation of data collection. RTJ, DEG and JA conceived of the analytic plan with input from TBN, SAL and AP. RTJ conducted the quantitative analysis. RTJ lead the writing of the manuscript, with contributions, review and approval from all authors. All authors read and approved the final manuscript. RTJ and JA were responsible for the decision to submit the manuscript. RTJ is acting as guarantor.

**Funding** This study was funded by the National Institutes of Health (grant R01 MH110186) and by the National Institute of Allergy and Infectious Diseases, the National Institute of Mental Health and the National Institute on Drug Abuse (awards UM1 AI068619 (HPTN Leadership and Operations Center), UM1AI068617 (HIV Prevention Trials Network Statistical and Data Management Center) and UM1AI068613 (HIV Prevention Trials Network Laboratory Center)). The content is solely the responsibility of the authors and does not necessarily represent the official views of the National Institutes of Health. This work was also supported by the National Institute of Mental Health (grant R01MH087118) and the Carolina Population Center (grant P2C HD050924). Additional support was provided by the Division of Intramural Research, National Institute of Allergy and Infectious Diseases. The MRC/Wits Rural Public Health and Health Transitions Research Unit and Agincourt Health and Socio-Demographic Surveillance System, a node of the South African Population Research Infrastructure Network (SAPRIN), have been supported by the National Department of Science and Innovation, the University of the Witwatersrand, and the Medical Research Council, South Africa, and by the Wellcome Trust, United Kingdom (grants 058893/Z/99/A, 069683/Z/02/Z, 085477/Z/08/Z and 085477/B/08/Z).

**Competing interests** None declared.

**Patient and public involvement** Patients and/or the public were involved in the design, or conduct, or reporting, or dissemination plans of this research. Refer to the Methods section for further details.

**Patient consent for publication** Not applicable.

**Ethics approval** This study involves human participants and was approved. Ethical approval for this study was obtained by from the University of North Carolina at Chapel Hill (#10-1868), the University of the Witwatersrand Human Research Ethics Committee (#101012) and the Mpumalanga Province's Health Research and Ethics Committee. Approval for analytical work was also obtained by the University of California, Berkeley, and the University of California, San Francisco. No reference number or ID for ethical approval from the University of California, Berkeley, the University of California, San Francisco, or the Mpumalanga Province's Health Research and Ethics Committee is available. Participants gave informed consent to participate in the study before taking part.

**Provenance and peer review** Not commissioned; externally peer reviewed.

**Data availability statement** No data are available. Data access to HPTN 068 is managed by FHI360. Data requests for de-identified data can be made by contacting Erica Hamilton at EHamilton@fhi360.org

ORCID iD
Ruvani T Jayaweera http://orcid.org/0000-0003-0609-9892

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
