## [Reviewer comments · BMJ Open]

ARTICLE DETAILS

TITLE (PROVISIONAL)	Associations between WASH-related violence and depressive symptoms in adolescent girls and young women in South Africa (HPTN 068): a cross-sectional analysis
AUTHORS	Jayaweera, Ruvani T.; Goin, Dana; Twine, Rhian; Neilands, Torsten; Wagner, Ryan; Lippman, Sheri; Kahn, Kathleen; Pettifor, Audrey; Ahern, Jennifer

VERSION 1 – REVIEW

REVIEWER	Corinne Wallace University of Saskatchewan College of Arts and Science, Department of Geography and Planning
REVIEW RETURNED	25-Feb-2022

GENERAL COMMENTS	The manuscript represents a well written description of how experiences and fear of WaSH-related violence may be related to adverse mental health. However, greater description and explanation of methods would be useful. Depending on considerations and specific methods, the manuscript may require more than minor revisions through a re-analysis of data. Page 6, lines 30-33 needs references for the confounders as well as a statement on confounders identified in the literature that could not be explored due to lack of appropriate data in the survey that was undertaken. For example, recent catastrophic events and water insecurity in itself have been identified as causes of adverse mental health. Further, while you used deciles of household assets, Workman and Ureksoy spoke to changing household status/assets, suggesting that perhaps differences between surveys may be at least as useful to examine as the point data used currently (Workman, C. L., & Ureksoy, H. (2017). Water insecurity in a syndemic context: Understanding the psycho-emotional stress of water insecurity in Lesotho, Africa. Social Science & Medicine, 179, 52-60). A second clarification regarding the methods is whether you separated out those who indicated experience versus fear so that you were looking at distinct groups of experience and fear in your analyses (as opposed to both)? This would be necessary to determine any differences that actual experience versus fear of potential experience manifest in terms of adverse mental health impacts. Presumably if an individual has experienced violence, they will fear re-occurrence. This would be a useful piece of analysis and would ensure that experience is not a confounder in the fear analyses and vice versa. Your table 1 suggests that there may be significant overlap, which would not allow you to determine whether
---

	it was the experience or fear (or both) that contributes to depression. Specific Edits: Pg 4, Line 5-6: Please use 2020 data from the 2021 JMP report. Pg 4, Line 52: "WaSH facilities on mental health" as opposed to and? Pg 5: Summary box - perhaps include a statement that women and girls have different daily WaSH needs compared to men and boys? Pg 5, Line 19: WaSH-related violence? Pg 6, Line3: please reference and include a sentence on why this method (ACASI) is preferred/performs better over in person interviews. Pg 7, Line 14: Please explain rationale for not reporting back/disseminating results back to the study region. Pg 7, line 31: last recorded visit to the toilet facility? why not over life course to the point in time of data collection? Please clarify and correct as necessary. Pg 7, Line 36: what were the types of water and sanitation facilities used? Did these correlate in any way with socio-economic status? or with likelihood of experiencing / fear of violence? Pg 11, Line 35: you are missing some of the local/household water security literature, which includes authors such as Wutich, Workman, Boateng.
--	--

REVIEWER	Sheillah Simiyu African Population and Health Research Center
REVIEW RETURNED	10-Mar-2022

GENERAL COMMENTS	I have asked the authors to check and update their references. The manuscript can also be reviewed for minor editorial mistakes Comments to the author: This paper presents findings on violence, an depression among young girls and women in South Africa. The study used secondary data from the HPTN 068 cohort study. Comments are outlined below Introduction, page 3 Line 6-8: "Much of the existing research on water, sanitation, and hygiene (WASH) has focused on pathogen-related risks to health or long-term health impacts of infectious diseases" Comment: This statement is not entirely true, as it seems to suggest that most of the WASH research has focused on pathogen related risks to health. You probably meant to say that much of the health focused existing research on WASH Line 10-11: "The experience of physical or sexual violence when traveling to or using water and sanitation facilities (WASH-related violence) is an important and under-examined WASH health issue" Comment: Some research on this topic has been done especially in India. You may want to quote these already existing studies Line 21-25: "A literature review found many examples from humanitarian case studies and practitioner reports that document the heightened risk of sexual violence faced when using communal latrines or practicing open defecation, particularly at night, as well as the heightened risk of sexual violence when traveling to collect water" Again, there are studies from India on this topic, you may want to add that literature here Methods: Perhaps since no indicators of hygiene were measured, may find it easier to revise your reporting, including the title to 'water and
---

	sanitation' rather than 'WASH' Results: Were there any efforts to compare the experiences between the trial arms? Table 1 provides results across the arms. Line 55: 'identifies sanitation and hygiene facilities as a structural/built environment determinant of violence against women and poor mental health outcomes' Comment: Since you did not include hygiene, you way want to state that 'this identifies water and sanitation facilities as ... Comment: In General, there are studies on water and sanitation, violence and mental health; rather than repeatedly stating that there are no studies, you may rather want to state the gaps in the existent studies, even if they are few Page 10 line 16: You state: consider the built environment, and specifically the design of water Comment: Your study did not consider design, rather considered access. You may want to revise the statement accordingly Page 10, line 29: You state: "Thus, addressing built environment determinants of depression like WASH-related violence may help improve mental health outcomes as well as other forms of well-being among adolescent girls..." Comment: Is violence a determinant of the built environment? PAGE 10 LINE 35: "Beyond direct experiences of violence, qualitative findings in the WASH literature have highlighted the role that fear or worry about experiencing violence—which in itself is a form of violence—may play in poor mental health outcomes" Comment: Please include a reference Comment: Overall, the findings are rather obvious, that it is expected that violence will lead to depression, and that fear of, or worry of experiencing violence can lead to depression. Were there any unexpected, or underreported findings that can supplement the findings?
--	--

VERSION 1 – AUTHOR RESPONSE

Reviewer: 1

Dr. Corinne Wallace, McMaster University

Comments to the Author:

The manuscript represents a well written description of how experiences and fear of WaSH-related violence may be related to adverse mental health.

However, greater description and explanation of methods would be useful. Depending on considerations and specific methods, the manuscript may require more than minor revisions through a re-analysis of data.

Response: Thank you for your extremely helpful comments and suggestions, we have revised our analysis based on your feedback and have updated our manuscript accordingly (further details provided below).

Page 6, lines 30-33 needs references for the confounders as well as a statement on confounders identified in the literature that could not be explored due to lack of appropriate data in the survey that was undertaken. For example, recent catastrophic events and water insecurity in itself have been identified as causes of adverse mental health. Further, while you used deciles of household assets, Workman and Ureksoy spoke to changing household status/assets, suggesting that perhaps differences between surveys may be at least as useful to examine as the point data used currently (Workman, C. L., & Ureksoy, H. (2017). Water insecurity in a syndemic context: Understanding the psycho-emotional stress of water insecurity in Lesotho, Africa. *Social Science & Medicine*, 179, 52-60).

Response: Thank you for these helpful comments and helpful references; while we lack information on water insecurity in our dataset, we hope that given the overlap in recent household food insecurity and water insecurity, controlling for food insecurity may at least partially control for this. In light of your suggestion, we additionally included recent household catastrophic events as a covariate in our models, as well as household water source and toilet type. We have added references to our statement on selection of confounders in the methods section, as well as a statement in the discussion acknowledging confounders we were unable to control for as a limitation.

This paragraph in the methods section now reads: “We used a directed-acyclic graph to identify a minimally sufficient set of literature-based confounders available in our study. We identified the following sociodemographic covariates of interest: age at time of survey[11, 12, 26], maternal and paternal education[26], orphan status[26], household food insecurity in the past 30 days[11, 18, 27], decile of household capita consumption[12, 26], and any negative events experienced by the household in the past 12 months[18, 27] (assessed via the household survey). Negative events reported in the household survey included experiences such as death or serious illness of a household member, loss of livestock or crop failure, job loss or loss of government grants, or loss or destruction of property. We also controlled for toilet type[9, 11] and household water source[18].”

We have added the following sentence to the discussion: “While we were unable to control for distance to toilet or water source, we hypothesize that the potential confounding effects of these variables are at least partially accounted for by toilet type and water source.”

A second clarification regarding the methods is whether you separated out those who indicated experience versus fear so that you were looking at distinct groups of experience and fear in your analyses (as opposed to both)? This would be necessary to determine any differences that actual experience versus fear of potential experience manifest in terms of adverse mental health impacts. Presumably if an individual has experienced violence, they will fear re-occurrence. This would be a useful piece of analysis and would ensure that experience is not a confounder in the fear analyses and vice versa. Your table 1 suggests that there may be significant overlap, which would not allow you to determine whether it was the experience or fear (or both) that contributes to depression.

Response: Thank you for this helpful comment. As the data are cross-sectional and assess prior direct experience of violence and prior worry about experiencing violence, we initially did not include the other measures of WASH-related violence in models assessing the associations between each individual exposure and depressive symptoms, as we were concerned about the temporality of these measures and the possible blocking of effects.

However, based on your feedback and a re-examination of our data, we agree that prior experience of violence, as measured in this study, likely precedes reporting of fear of violence, and can be considered a plausible confounder of the relationship between fear of violence and depressive symptoms. In particular, this allows us to examine the role of fear as distinct from its overlap with experiences of violence. As a result, we have included experience of violence when collecting water

as a confounder in the model assessing the relationship between fear of violence when collecting water and depressive symptoms, and have included experience of violence when using the toilet as a confounder in the relationship between fear of violence when using the toilet and depressive symptoms. Given our perspective that the most plausible ordering between these exposures is that experience of violence precedes fear of violence (as measured in this study), we do not include fear of violence in the models exploring experience of violence as the exposure of interest, as doing so would be controlling for a variable on the causal pathway.

The analysis section now reads, “We used g-computation to calculate the predicted marginal risk difference of depression and the predicted marginal prevalence ratio of depression associated with each individual domain of WASH-related violence. All models adjusted for age at time of visit, maternal education, paternal education, orphan status, household food insecurity, decile of household capita consumption, negative household experiences, and trial arm, and accounted for clustering by village. Models assessing fear of violence when collecting water and fear of violence when using the toilet additionally adjusted for prior experience of violence when collecting water and when using the toilet, respectively. Models assessing experience or fear of violence when collecting water additionally adjusted for household water source; models assessing experience or fear of violence when using the toilet additionally adjusted for household toilet type.”

We have updated the results throughout the paper and in the tables accordingly.

Specific Edits:

Pg 4, Line 5-6: Please use 2020 data from the 2021 JMP report.

Response: We have updated the data and reference. The sentence now reads, “Despite long-standing global efforts, approximately two billion people around the world lack access to safe water and 3.6 billion lack access to safely managed sanitation in 2020.”

Pg 4, Line 52: “WaSH facilities on mental health” as opposed to and?

Response: We have corrected this typo.

Pg 5: Summary box – perhaps include a statement that women and girls have different daily WaSH needs compared to men and boys?

Response: Per request from the editors, we have removed this summary box and corresponding text.

Pg 5, Line 19: WaSH-related violence?

Response: Per request from the editors, we have removed this summary box and corresponding text.

Pg 6, Line3: please reference and include a sentence on why this method (ACASI) is preferred/performs better over in person interviews.

Response: We have revised this sentence in the methods to now read, “Sensitive items, such as sexual behavior and mental health, were completed by the participant themselves via Audio Computer Assisted Self Interview (ACASI), where participants listen to questions and response categories through headphones and select their responses. Prior research has found higher reporting of sensitive issues via ACASI as compared to interviewer-administered surveys.[21]”

Pg 7, Line 14: Please explain rationale for not reporting back/disseminating results back to the study region.

Response: Apologies, this section was incomplete. There has been substantial community engagement and involvement with regards to the overall trial and study. We have amended this section in the manuscript to reflect this. The section now reads: “All studies, including the HPTN 068 trial, conducted in the AHDSS study site receive permission to undertake research activities from a forum comprised of community and village leaders. Findings from the main trial were communicated to the community via community meetings and factsheets. Additional details on community involvement are available via the study site’s Public Engagement Office. (https://www.agincourt.co.za/?page_id=1913)

Pg 7, line 31: last recorded visit to the toilet facility? Why not over life course to the point in time of data collection? Please clarify and correct as necessary.

Response: Last recorded visit referred to the study visit (visit 3) included in this study; exposure to violence was assessed over the life course to the point in time of data collection. We removed this phrase from the sentence to aid in clarity, sentence now reads: “Directly experiencing violence when using the toilet was reported by 15.1% of participants; experiencing violence when collecting water was reported by 17.1%”

Pg 7, Line 36: what were the types of water and sanitation facilities used? Did these correlate in any way with socio-economic status? or with likelihood of experiencing / fear of violence?

Response: We have added the household water source and toilet type to Table 1 and included it as a covariate in our models. Water source and toilet type was associated with depression; water source was also associated with any direct experiences of violence (see Table 1 footnotes).

Pg 11, Line 35: you are missing some of the local/household water security literature, which includes authors such as Wutich, Workman, Boateng.

Response: Thank you for this suggestion. We have updated our references to incorporate findings from this robust literature throughout the paper, and have also added a sentence to the Introduction highlighting the incorporation of worries about violence into measures of water insecurity and its associations with mental health: “However, new measures of water[15] and sanitation[16] insecurity include concerns about violence as a domain in these measures, and there is a growing body of research linking water and sanitation insecurity with poor mental health outcomes.[17-19]”

Reviewer: 2

Dr. Sheillah Simiyu, African Population and Health Research Center

Comments to the Author:

***** Please find additional comments from this reviewer in the attached file *****

I have asked the authors to check and update their references. The manuscript can also be reviewed for minor editorial mistakes

Response: Thank you for your helpful comments and suggestions. We have reviewed the literature and updated our references. To respond to the comments from Reviewer 1, we also re-ran our analyses accounting for additional covariates, and these results have been updated throughout the paper (though this re-analysis did not change our overall findings).

Comments to the author:

This paper presents findings on violence, an depression among young girls and women in South Africa.

The study used secondary data from the HPTN 068 cohort study. Comments are outlined below

Introduction, page 3

Line 6-8: “Much of the existing research on water, sanitation, and hygiene (WASH) has focused on pathogen-related risks to health or long-term health impacts of infectious diseases”

Comment: This statement is not entirely true, as it seems to suggest that most of the WASH research has focused on pathogen related risks to health. You probably meant to say that much of the *health focused existing* research on WASH

Response: Thank you, we have amended this statement to read, “Much of the existing **public health** research on water, sanitation, and hygiene (WASH) has focused on pathogen-related risks to health or long-term health impacts of infectious diseases.”

Line 10-11: “The experience of physical or sexual violence when traveling to or using water and sanitation facilities (WASH-related violence) is an important and under-examined WASH health issue” Comment: Some research on this topic has been done especially in India. You may want to quote these already existing studies

Response: Thank you – in the original paper, we cite key literature from India, including work from Jadhav, Khanna, and Sathoo. In our revision, we have incorporated additional research from India (as well as other countries), particularly work from Caruso and Kulkarni. We have revised our first paragraph to now say, “While recent studies have noted the increased vulnerability to violence that women and girls face when meeting their daily WASH needs[3, 4], the experience of physical or sexual violence when traveling to or using water and sanitation facilities (WASH-related violence) remains a under-examined issue.” The third and fourth paragraphs of the introduction provide additional references and details from the existing literature.

Line 21-25: “A literature review found many examples from humanitarian case studies and practitioner reports that document the heightened risk of sexual violence faced when using communal latrines or practicing open defecation, particularly at night, as well as the heightened risk of sexual violence when traveling to collect water” Again, there are studies from India on this topic, you may want to add that literature here.

Response: Thank you – we have highlighted the studies from India in the subsequent paragraph, which reads: “A growing body of research has further elaborated the ways in which shared water and sanitation facilities are a potential site of violence against women.[5] Qualitative research from Uganda, Kenya, and India have found that fear of sexual or physical assault and lack of privacy as driving factors in women’s sanitation practices and an important source of anxiety and stress.[3, 6-8] Quantitative evidence on the linkages between WASH and experiences of violence from India,[9, 10] South Africa,[11] and Kenya[12] have found that women who use open defecation or shared toilet facilities have higher odds of reporting non-partner sexual violence when compared to women with a household toilet. An analysis of Demographic and Health survey data from 25 African countries found that unimproved water and sanitation facilities was associated with higher odds of experiencing intimate partner violence.[13] An ecological study found a positive association between the number of sanitation facilities and the rate of reported sexual assaults in a township in South Africa.[14] However, none of these studies assessed whether women experienced violence while they were accessing or using WASH facilities.”

Methods:

Perhaps since no indicators of hygiene were measured, may find it easier to revise your reporting, including the title to ‘water and sanitation’ rather than ‘WASH’

Response: While we acknowledge that we do not have specific measures about experiences of violence when engaging in hygiene activities specifically, as access to water and toilet facilities has direct implications for household and individual’s ability to meet their hygiene needs, particularly in the context of menstrual hygiene.

Results:

Were there any efforts to compare the experiences between the trial arms? Table 1 provides results across the arms.

Response: We controlled for trial arm in the analysis as a potential confounder, and is including as a covariate in the final models. In Chi-Square tests of independence, trial arm was not related to the outcome (depressive symptoms).

Line 55: ‘identifies sanitation and hygiene facilities as a structural/built environment determinant of violence against women and poor mental health outcomes’ Comment: Since you did not include hygiene, you way want to state that ‘*this identifies water and sanitation facilities as ...*

Response: Thank you, we have revised this sentence to now read, “This analysis builds upon previous qualitative work and practitioner reports, and identifies water and sanitation facilities as a structural/built environment determinant of violence against women and poor mental health outcomes.”

Comment: In General, there are studies on water and sanitation, violence and mental health; rather than repeatedly stating that there are no studies, you may rather want to state the gaps in the existent studies, even if they are few

Response: To our knowledge, we have not identified any studies that have quantitatively assess the burden of direct experiences of WASH-related violence on mental health outcomes. We agree that this work builds on important prior work that has highlighted the linkages between water, sanitation, violence, and mental health. We have incorporated additional literature at the recommendations of both reviewers, and attenuated our statements throughout the paper to make this more clear.

**Page 10 line 16: You state: consider the built environment, and specifically the *design* of water
Comment: Your study did not consider design, rather considered access. You may want to revise the statement accordingly.**

Response: Thank you, statement now reads: “Indeed, much of the focus on prevention of violence against women is centered on individual, household, and community-level risk factors; our findings further support calls made by others[7] to consider the role of the built environment, and specifically **access to** water and sanitation facilities, as an important factor that may influence non-partner violence against women.”

Page 10, line 29: You state: “Thus, addressing built environment determinants of depression like WASHrelated violence may help improve mental health outcomes as well as other forms of well-being among adolescent girls...” Comment: Is violence a determinant of the built environment?

Response: Thank you for this comment – violence is not a determinant of the built environment, but the built environment (design and access to WASH facilities) is a determinant of exposure to violence

and consequent additional downstream effects, such as depressive symptoms as explored in this paper. We have reworded this sentence to make this clear, sentence now reads: “Thus, addressing determinants of depression **related to the built environment**, such as **access to WASH facilities and consequent exposure to WASH-related violence**, may help improve mental health outcomes as well as other forms of well-being among adolescent girls and young women.”

PAGE 10 LINE 35:

“Beyond direct experiences of violence, qualitative findings in the WASH literature have highlighted the role that fear or worry about experiencing violence—which in itself is a form of violence—may play in poor mental health outcomes” Comment: Please include a reference

Response: This sentence summarizes the references detailed in the rest of the paragraph. We have revised this sentence for clarity: “Beyond direct experiences of violence, findings in the WASH literature, **detailed below**, have highlighted the role that fear or worry about experiencing violence—which itself is a form of violence—may play in poor mental health outcomes.”

Comment: Overall, the findings are rather obvious, that it is expected that violence will lead to depression, and that fear of, or worry of experiencing violence can lead to depression. Were there any unexpected, or underreported findings that can supplement the findings?

Response: While the findings are in the direction one might expect, such that experience of violence and fear of violence is associated with higher prevalence of depressive symptoms; we believe that these findings are novel and important and this is the first attempt to quantify the difference in magnitude of this prevalence based on violence experiences. The magnitudes of the associations are substantial, particularly for experienced violence when using the toilet, where those who experienced violence had a 18.1 percentage point higher absolute prevalence of depressive symptoms as compared to those who did not experience violence when using the toilet. Per comments from the other reviewer, we adjusted our modeling strategy, which highlights that even when controlling for prior experience of violence, worry about violence when using the toilet remains a salient independent risk factor for depression.

VERSION 2 – REVIEW

REVIEWER	Corinne Wallace University of Saskatchewan College of Arts and Science, Department of Geography and Planning
REVIEW RETURNED	10-Jun-2022
GENERAL COMMENTS	Thank you - this is much improved and a welcome addition to the literature. You may wish to alter the sentence "... additionally, the fear of experiencing violence when using the toilet, even when controlling for prior experience of experiencing violence when using the toilet..." to "... additionally, the fear of experiencing violence when using the toilet, even when controlling for prior experience of violence when using the toilet..."